**Data Availability Statement:** Data on footpaths, roads, community health sites, and building are

# Combining OpenStreetMap mapping and route optimization algorithms to inform the delivery of community health interventions at the last mile

**Mauricianot Randriamihaja**[1,2,3]*, **Felana Angella Ihantamalala**[1,4], **Feno H. Rafenoarimalala**[1], **Karen E. Finnegan**[1,4], **Luc Rakotonirina**[1], **Benedicte Razafinjato**[1], **Matthew H. Bonds**[1,4], **Michelle V. Evans**[1,4], **Andres Garchitorena**[1,3]

**1** NGO Pivot, Ranomafana, Ifanadiana, Madagascar, **2** ED 168 CBS2, University of Montpellier, Montpellier, France, **3** MIVEGEC, University of Montpellier, CNRS, IRD, Montpellier, France, **4** Department of Global Health and Social Medicine, Harvard Medical School, Boston, Massachusetts, United States of America

* mauricianothr@gmail.com

## Abstract

Community health programs are gaining relevance within national health systems and becoming inherently more complex. To ensure that community health programs lead to equitable geographic access to care, the WHO recommends adapting the target population and workload of community health workers (CHWs) according to the local geographic context and population size of the communities they serve. Geographic optimization could be particularly beneficial for those activities that require CHWs to visit households door-to-door for last mile delivery of care. The goal of this study was to demonstrate how geographic optimization can be applied to inform community health programs in rural areas of the developing world. We developed a decision-making tool based on *OpenStreetMap* mapping and route optimization algorithms in order to inform the micro-planning and implementation of two kinds of community health interventions requiring door-to-door delivery: mass distribution campaigns and proactive community case management (proCCM) programs. We applied the Vehicle Routing Problem with Time Windows (VRPTW) algorithm to optimize the on-foot routes that CHWs take to visit households in their catchment, using a geographic dataset obtained from mapping on *OpenStreetMap* comprising over 100,000 buildings and 20,000 km of footpaths in the rural district of Ifanadiana, Madagascar. We found that personnel-day requirements ranged from less than 15 to over 60 per CHW catchment for mass distribution campaigns, and from less than 5 to over 20 for proCCM programs, assuming 1 visit per month. To illustrate how these VRPTW algorithms can be used by operational teams, we developed an "e-health" platform to visualize resource requirements, CHW optimal schedules and itineraries according to customizable intervention designs and hypotheses. Further development and scale-up of these tools could help optimize community health programs and other last mile delivery activities, in line with WHO recommendations, linking a new era of big data analytics with the most basic forms of frontline care in resource poor areas.

available on OpenStreetMap (https://www.openstreetmap.org) and the Shiny app (http://research.pivot-dashboard.org/) contains the results data.

**Funding:** This work was supported by the Agence Nationale de la Recherche (SMALLER project ANR-19-CE36-0001-01 to AG and MVE), the French Development Agency (AFD) as part of the PREACTS program (AFRICAM project to AG and MR) and internal funding from Pivot (to AG and MR). The funders had no role in study design, data collection and analysis, decision to publish, or preparation of the manuscript.

**Competing interests:** The authors have declared that no competing interests exist.

## Author summary

Community health programs play a critical role in providing equitable health care, especially in remote and underserved areas. However, these programs are often complex and difficult to implement effectively. This study explores how geographic optimization can improve the delivery of community health interventions, ensuring that community health workers (CHWs) can efficiently reach every household. By integrating OpenStreetMap data into route optimization algorithms, we developed a decision support tool to streamline the planning and execution of door-to-door health services. In the rural district of Ifanadiana, Madagascar, this tool was used to optimize CHW routes for mass distribution campaigns and proactive community case management (proCCM). The results showed that these optimizations could significantly reduce the time and resources required of CHWs. To facilitate its practical application, we created an "e-health" platform that displays resource needs and CHW routes for each activity. This approach promises to improve last mile health care delivery, in alignment with WHO guidelines, and use advanced data analytics to improve frontline health care in resource-limited settings.

## Introduction

Nearly half of the world's population still lacks access to essential health services [1]. As countries strive to make progress towards universal health coverage and to meet the health-related Sustainable Development Goals, community health systems have played a crucial role in reducing geographic barriers to primary care and ensuring the delivery of health interventions at the last mile [2,3]. For instance, the expansion of integrated community case management (iCCM) programs across low- and middle-income countries, where community health workers (CHWs) diagnose and treat common illnesses for children under 5 years of age, has contributed to improving healthcare access and has been associated with reductions in mortality rates for this age group [4–7]. In addition to iCCM, CHWs play a key role in a variety of health activities at the community level, including health awareness campaigns, patient monitoring, health promotion, surveillance, and providing high-impact interventions aimed at reducing maternal and child mortality, such as mass drug administration or bednet distribution. More recently, CHWs have been increasingly involved in proactive community case management programs (proCCM), which involve systematically visiting all the households in a catchment on a regular basis (e.g. once per month) to provide health services to those in need. These approaches have been associated with important health benefits such as reductions in malaria prevalence and have been credited for reducing child mortality [5,8–11].

As community health programs gain relevance within national health systems, they are becoming inherently more complex, with larger scopes of work and target populations [3,12]. There are multiple areas amenable to optimization. International guidelines include recommendations focused on their professionalization and integration, including training, supportive supervision, remuneration, strengthening supply chains, and community mobilization [3]. To ensure that community health programs lead to equitable geographic access to care, the WHO recommends adjusting the target population and workload of CHWs according to the local geographic context and population size of the communities they serve [3]. Indeed, geographic barriers persist even in the delivery of community health interventions that are meant to solve them, such as iCCM [13] or vaccination campaigns [14]. Geographic optimization can be particularly beneficial for those activities that require CHWs to visit hundreds of

households for last mile delivery of care, whether during routine visits such as in proCCM and active surveillance activities, or during mass distribution campaigns (e.g. bednets, chemoprophylaxis, immunizations). Thus, decision-making tools aimed at route optimization could be a valuable resource for program managers and CHWs alike in guiding the micro-planning and implementation of door-to-door community health delivery interventions [15].

Route optimization algorithms have been around for more than 50 years and they have transformed delivery operations for products and services, with applications in sectors as diverse as package delivery, supply chain management, garbage collection, and home health-care services [15,16]. These algorithms are generally referred to as the Vehicle Routing Problem (VRP) and the Traveling Salesman Problem (TSP), and among their many variations they can incorporate time window constraints (VRPTW and TSPTW) to account for work schedules [16]. Both the TSPTW and VRPTW algorithms share the same route optimization principle. However, VRPTW is capable of considering situations where multiple vehicles (or personnel) visit various locations before returning to the initial point, while TSPTW is limited to a single visiting vehicle (or person) [17]. There have been dozens of studies where these algorithms were applied to home health care service delivery in Europe and North America for nurses doing home visits [16]. There is no equivalent literature on the application to community health delivery in low and middle income countries (LMICs), even though they share similar features, including time constraints and the need to cover households over geographically extensive areas.

An important obstacle for the application of routing algorithms is the lack of publicly available geographic data at the granular level of individual households and footpaths in rural areas of the developing world where community health programs are essential. Indeed, the completeness of free, open geographic databases such as *OpenStreetMap* (OSM), which is updated and maintained by a community of volunteers, is still very low and unequally distributed across the world. For instance, map completeness of urban areas in regions outside of Europe and North America ranges from 10% to 30%, and completeness is inversely correlated with the population size of cities [18]. This suggests that map completeness may be particularly low in rural areas of LMICs, where community healthcare programs can be most beneficial.

Using a rural district of Madagascar as a case study, we demonstrate here how OSM data and VRPTW algorithms can be combined to develop decision-making tools for the geographic optimization of community health activities and last mile interventions that require door-to-door delivery. We leveraged a complete *OpenStreetMap* mapping dataset of Ifanadiana district, in southeastern Madagascar, which was done to support community health activities during a health system strengthening initiative and resulted in over 100,000 buildings and 20,000 km of footpaths mapped [19]. The tool is based on the VRPTW optimization algorithm and was conceived to guide the micro-planning and implementation of two kinds of community health interventions requiring door-to-door delivery: mass distribution campaigns and proCCM programs. Use of this tool could help maximize geographic coverage while minimizing resources and costs associated with the implementation of these last mile delivery interventions.

## Materials and methods

### Study area

This study was conducted in Ifanadiana district, located in the Vatovavy region of southeastern Madagascar. It covers a total area of 3,975 km$^2$ and has a population of approximately 200,000 inhabitants. The district comprises 15 communes (an administrative unit with an average total of 13,000 people) and 195 fokontany (local administrative unit, comprising multiple villages with an average total of approximately 1000 people), with the majority accessible only by foot

due to its mountainous terrain. The district is connected by a paved national road from east to west, while nearly all remaining routes are unpaved. The healthcare system in the district includes a reference hospital (CHRD II), 15 Level II primary health centers (PHCs) (one per commune), and 6 Level I PHCs, which provide basic services in communes that are more geographically dispersed. In line with national community health policies, each fokontany possesses a community health site (CHS)—a physical structure where two community health workers (CHWs) provide clinical consultations such as for iCCM and malnutrition, and family planning [20]. iCCM involves diagnosis and treatment for diarrhea, respiratory illness, and malaria, for children under-five. In addition, CHWs frequently take part in mass distribution campaigns, which are typically done door-to-door, such as mosquito-net distribution, mass drug administration, or childhood vaccination. Importantly, while there is only one CHS per fokontany (*i.e.*, the catchment area of CHWs), the sizes of the fokontany in the district vary by orders of magnitude, ranging from 2 to 139 km$^2$.

Since 2014, a collaboration between the Ministry of Public Health and the non-governmental health organization Pivot has aimed to strengthen the public health system at all levels of care (hospital, health centers and community health) and enhance the quality of care [21]. For community health, Pivot supports the activities of 180 out of the 390 CHWs, covering half of the district. These CHWs are supervised by 14 community health supervisors to ensure proper training and equipment provision, enabling them to deliver quality care to their communities. However, access to healthcare at the community level remains problematic due to geographical barriers. Less than one quarter of households live in the immediate vicinity (defined as within 1 km) of a CHS [19], and distance to the CHS is associated with a decrease in utilization by over 25% per kilometer [13]. To further reduce geographic barriers, a proCCM program was piloted in one commune, where CHWs conducted home visits to every household in their catchment at least once per month, allowing them to consult, treat children, do door-to-door sensitization and follow-up on unwell individuals within their communities [6]. The financial and human resources necessary for the scale-up of such a program depend on their clinical workflows, population size, and the specific geographic context of each fokontany in the district.

## Data collection

The foundation for our work was a comprehensive geographic information system of Ifanadiana District, obtained via mapping on the OSM platform. The full details of this mapping exercise are reported in Ihantamalala et al. 2020. In brief, digital mapping was carried out in collaboration with the Humanitarian *OpenStreetMap* Team (HOT), involving the division of the district into more manageable 1 km$^2$ tiles [22]. The mapping involved overlaying geographic entities (roads, paths, households) onto high-resolution satellite imagery via photo-interpretation. The process was carried out in two steps: initially, one user mapped all geographic entities, followed by another user validating the mapping to ensure data quality. To obtain this data, we downloaded OSM information using the "QuickOSM–Overpass API" extension of QGIS [23], which included an extensive network of 20,000 km of footpaths, 192 km of roads, 108,000 buildings, and 195 community sites [19]. We also obtained administrative boundaries for the 15 communes and 195 fokontany in the district [24]. We used spatial joining techniques to link downloaded OSM information to administrative boundaries to create comprehensive geographic profiles for each fokontany, which is the unit of analysis of this study.

## Data analysis

We used the Google OR-Tools optimization VRPTW algorithm, version 9.1.9490 [25] to optimize CHW routes and home visits in each of the community health catchments (*i.e.*

fokontany). The algorithm was originally designed to optimize fleets of vehicles, such as delivery trucks, by creating a set of routes that minimize the travel time of the fleet of traveling units departing from a central node and visiting a number of clients. It adheres to specific time constraints within a given time window and ensures that each client is serviced exactly once without exceeding the fleet capacities. The details of this algorithm are described extensively in [26]. We applied the VRPTW algorithm for two scenarios, which correspond to two types of community health interventions requiring door-to-door delivery: mass distribution campaigns and proCCM programs. In both scenarios, the vehicle fleet was represented by survey teams or CHWs who walk along footpaths within a rural district (instead of driving along a road), the central node was the CHS, and the clients were the households or buildings to be visited. For each scenario, we defined a set of assumptions and parameters (e.g. % of buildings to visit, time per visit, number of visits, etc.) based on contextual knowledge of how these interventions are implemented in Ifanadiana. Before running the VRPTW algorithm, we obtained the shortest paths and corresponding travel time estimates between each pair of buildings (including the CHS) in a fokontany. This distance matrix was used as input data in the VRPTW algorithm to estimate optimal routes for CHWs to visit every household in the district and the associated resources in terms of person-days necessary to complete the work. Finally, we integrated the algorithm and these analyses into an e-health tool for operational use by local program managers and community health workers. Each of these steps is elaborated in the following sections.

## Scenario assumptions

We assumed that a CHW can work up to 8 hours per day over 20 working days in a month, with an average walking speed of 5 km/h [27]. While previous research in our District [19] and elsewhere [28] reveals that walking speed can vary according to local conditions in land cover, terrain, and climate, we made this choice for processing efficiency, as 5 km/h was the average walking speed in Ifanadiana [19]. For simplicity, both interventions assumed that a health worker starts their work at the CHS, then walks along paths to reach the households for visits. At the end of each day, CHWs return to the starting point (Fig 1). Given a total population of 200,000 and an average household size of approximately 5 individuals per household [29], we assumed that only 40% of the 108,000 buildings were permanently occupied households (Table 1), while the rest would be other types of buildings (e.g. administrative buildings, shops, storage spaces, secondary homes near fields, etc.). We randomly assigned a building's status (e.g. occupied vs. unoccupied) following the percentages above for each fokontany and ensured there was no spatial correlation in this sampling procedure. The scenarios differed primarily in the percentage of households to be visited, the frequency of visits, and the duration of each visit. This random allocation was used because the photointerpretation of satellite imagery used for mapping the District does not allow us to discriminate between permanently occupied households and unoccupied buildings used for other purposes. This information, which is necessary for program implementation, can be obtained via an initial census or through participatory approaches with local populations, but this was beyond the scope of our study which focused on the development of the optimization method.

For mass distribution campaigns (scenario one), we assumed that CHWs would have to visit every building in their catchment area, whether occupied or not, in order to obtain a complete enumeration and accurate numbers of households or individuals requiring the product delivered (e.g. bed nets, drugs, immunizations). We assumed that CHWs would spend 5 minutes at uninhabited buildings, and 30 minutes at occupied households to allow sufficient time to explain the campaign, distribute the products, and gather basic socio-demographic information for reporting (Table 1). The assumptions and results in this scenario are also applicable to

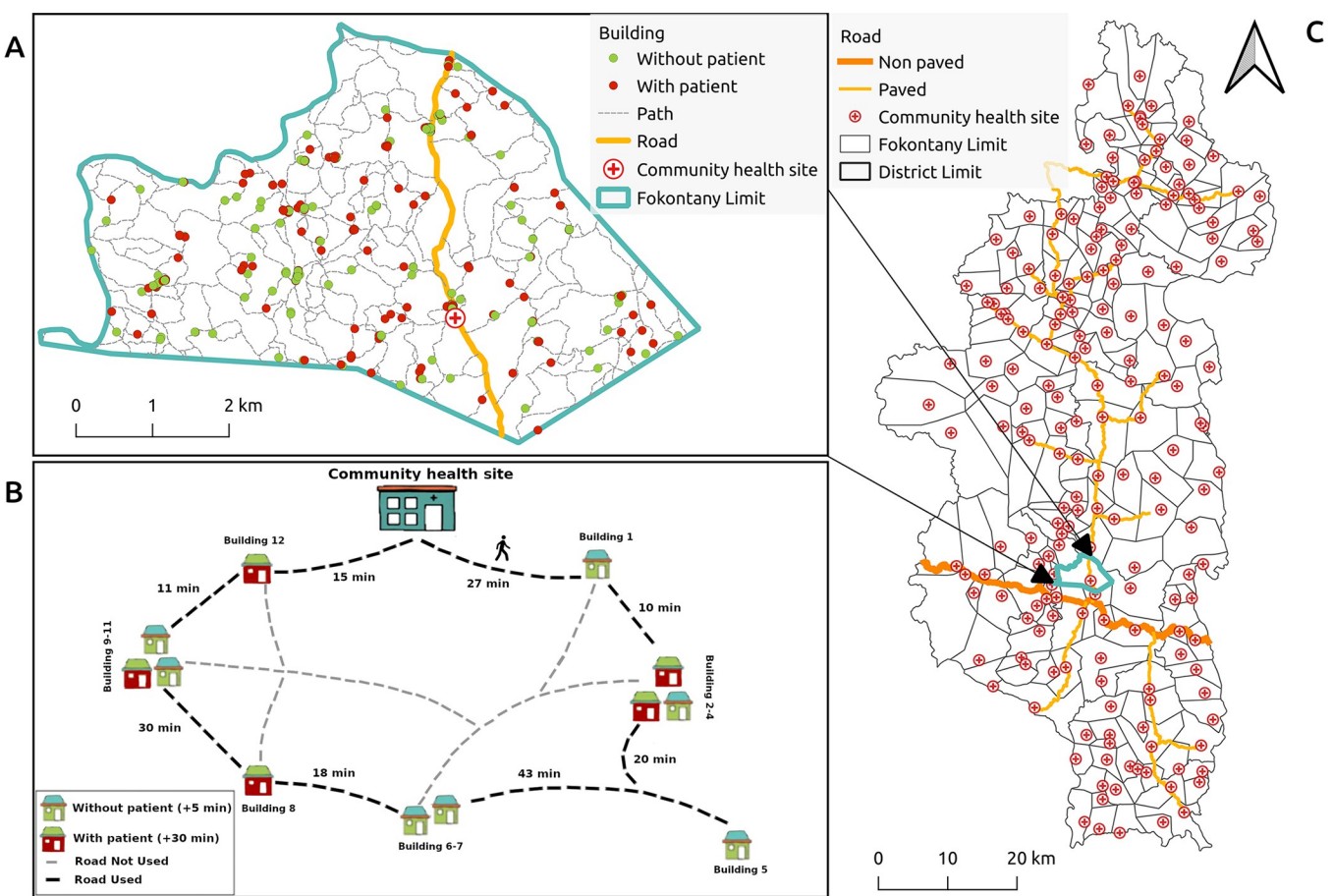

**Fig 1. Study area and conceptual diagram of algorithm implementation in one community health worker catchment.** A) Illustrative example of geographic information available for a community health worker catchment. B) Schematics of a CHW's daily route plan during proCCM, where household visits are conducted within a maximum time window of 8 hours to return to the same starting point. C) Map of Ifanadiana district highlighting the location of major roads and CHSs. Sources: HumData Madagascar (https://data.humdata.org/dataset/cod-ab-mdg) used for administrative boundaries, Contains information from OpenStreetMap and OpenStreetMap Foundation, which is made available under the Open Database License (https://www.openstreetmap.org/copyright).

situations where a census (full enumeration) of the catchment area is needed prior to conducting activities such as a proCCM program, a representative survey (e.g. demographic and health survey) or a national census.

For proCCM program implementation (scenario two), we assumed that CHWs would only visit the 40% of buildings assumed to be occupied households in scenario one. Of these occupied households, we assumed that 80% would not require the provision of care (diagnosis, treatment) or other time-consuming activities, while 20% of households would require an intervention by the health worker (Table 1). This is based on the assumption that CHWs provide care to children under 5 years; the percentage of children reporting an illness in the previous two weeks ranged from 20 to 30% in a population-representative cohort in Ifanadiana [30]. As with building occupation, we randomly assigned those 20% of households requiring an intervention within each fokontany. For each proCCM visit, we assumed a visitation time of 5 minutes if the household did not have a sick child, and 30 minutes if the household did. These visitation times are in accordance with average times witnessed through Pivot's own community health program during iCCM consultations. It is important to note that we modeled a simple proCCM scenario, where all inhabited households in the CHW catchment are

**Table 1. Model assumptions for mass distribution campaign and proCCM intervention scenarios.**

| Variables | Mass distribution campaign | proCCM program |
|---|---|---|
| Duration of work per day | 8 hours | |
| Number of working days per month | 20 days | |
| Average CHW walking speed | 5 kilometers per hour | |
| Starting and ending location | The CHS (or the village center, when CHS location not available) of the fokontany is the starting and ending point each day for a CHW conducting household visits. | |
| Percentage of inhabited households among buildings | 40%* of buildings are considered to be inhabited households | |
| Uninhabited household data | CHW visits all buildings, regardless of whether they are inhabited or not. | CHW visits only the 40%* of buildings considered as inhabited households |
| Characteristics of inhabited households | No difference among inhabited households | 20%* of inhabited buildings are considered households with patients (or sick individuals) and 80%* without patients |
| Duration of visit in each household | Inhabited households: 30 minutes | Households with patients: 30 minutes (range 10–120 minutes in sensitivity analyses) |
| | Uninhabited buildings: 5 minutes | Households without patients: 5 minutes (range 5–60 minutes in sensitivity analyses) |

* Households are randomly assigned each status, according to the percentages described above.

visited with a given frequency (1 to 4 times per month) without follow-up visits, and with constant visitation times so that our methods and results are broadly applicable. In reality, the task of CHWs implementing proCCM in the District of Ifanadiana is even more complex: they are responsible not just for treatment of sick children, but for follow-up visits of these children, referral of sick adults to PHCs, diagnostics for malaria and tuberculosis, follow-up for patients enrolled in malnutrition programs, and health education, all of which may exhibit seasonality. In addition, the variety and intensity of activities varies greatly from day-to-day and between districts. A detailed description of the proCCM program implemented in Ifanadiana District and its impact are available in Razafinjato et al. (2024) [6].

**Estimating optimal itineraries and determining the number of CHWs required for each intervention.** We implemented the default VRPTW algorithm developed by Google in Python 3.10 [25], using the reticulate package [31] for implementation with R software, version 4.3. The VRPTW algorithm uses four parameters: a travel time matrix (including the visit time per building), the departure and arrival locations, the time window for visits, and the number of CHWs [25]. The travel time matrix is a matrix of paired travel times between each and all destinations to be visited within a CHW catchment. We constructed this matrix using the Open Source Routing Machine (OSRM) tool which employs Dijkstra's routing algorithm to find the shortest path between two nodes in a network, applied to the complete set of buildings and paths available on OpenStreetMap for Ifanadiana District [32]. We used the R package "osrm" for implementation of OSRM with R software [33]. Via the OSRM routing algorithm, we obtained two outputs: a paired matrix of travel times between each building (including the CHS), and the shortest travel path between each pair of points. A separate travel time matrix was constructed for each fokontany and each scenario. Buildings in the dataset were randomly assigned the status of inhabited household or uninhabited in the mass distribution scenario, and the status of inhabited household with a patient or without a patient in the proCCM scenario according to probabilities described above. This vector of visit times was

integrated into the matrix of travel time for each building in the dataset prior to running the VRPTW algorithm. The departure and arrival locations were consistently set as the CHS (or village center for the fokontany without CHS) and we set the time window to an average work-day of 8 hours.

The algorithm requires appropriate starting parameters to converge on an optimal solution. We therefore estimated the remaining parameter, the number of CHWs, in two steps. We first estimated the total travel time required to complete all visits in a catchment without consider-ing a time window constraint and returning to the departure point upon completion. From this, we estimated the number of 8h-days it would take one CHW to complete that work, dividing the total travel time by 8h and rounding up to the nearest integer. Using this number as the starting parameter for the number of CHWs needed to complete the work in one 8h-day, we then added the time window constraint and ran the algorithm again. If the algorithm did not converge with this starting parameter, we iteratively incremented the number of CHWs by 1 and ran the optimization procedure again, continuing until the algorithm identi-fied a solution (S1 Fig). The outcome of the VRPTW algorithm is the optimal itineraries per CHW-day for each fokontany to cover the entire catchment, with details of the households to be visited, the order of visits, and paths to take. The total CHW-days obtained per fokontany thus represent both the number of days required for one CHW to complete the work and the number of CHWs required to complete the work in one day.

We further explored the sensitivity of our findings to our model assumptions for the proCCM scenario, and obtained estimates under scenarios with differing monthly frequency of visits to each household and differing visit times. We considered three monthly frequencies of visits to each household: one visit per month, one visit every two weeks (two visits per month), and one visit per week (four visits per month). We also explored a range of visit times for patient households (ranging from 10–120 minutes in intervals of 10) and non-patient households (ranging from 5–60 minutes in intervals of 5). We simulated each combination of visit times and visit frequencies once, resulting in 576 simulated scenarios. This approach enabled us to provide more flexible results for program managers, which can be used to adapt the design of proCCM programs (number of tasks to be done and time spent in each visit, number of visits, etc.) according to particular goals and available resources.

We also conducted sensitivity analyses on the randomization of the definition of occupied households vs. other building types for both scenarios. We created 100 new datasets of occu-pied household locations by randomly selecting 40% of the buildings to be permanently occu-pied. We then reran the optimization algorithm for the proCCM scenario, following the default model assumptions (Table 1) for each of these new datasets, and compared the number of personnel-days for each randomization with the initial estimate at both the fokontany and district level.

**Integrating VRPTW algorithms into an e-health tool.** We developed a web application using "R Shiny" for Ifanadiana district to illustrate how the results obtained with the method developed in this study can be used for decision-making, program planning and design, and implementation of field activities. The application can provide information for three use cases: i) mass distribution campaigns and household censuses, ii) proCCM programs and active sur-veillance, and iii) customized field visits. For mass distribution campaigns and censuses, users can estimate the number of days required to complete the work in each fokontany. Users can input the number of CHWs (or other health workforce) available per fokontany and modify default simulation parameters to observe their effects on the duration in terms of workdays. For pro-CCM programs, users can estimate the number of CHWs needed per fokontany based on the targeted number of visits per household per month, and they can adjust certain parameters according to their assumptions in terms of visit time to understand their impact

on the required number of community agents to manage the workload. However, the tool does not yet incorporate more complex schedules such as follow-up visits to sick patients or additional activities included in the proCCM program. Other programmatic needs for conducting field visits are addressed via customized field visits. Users can choose the households to visit either interactively on a map or by uploading their coordinates, and specify the number of available personnel and the time needed for each household visit. This function assists in identifying the optimal routes and calculating the required personnel-days for the specified route. Beyond these estimations, the application provides detailed geographic information on the specific routes and households to be visited per day for field personnel, either on an OSM or satellite background map.

It is important to note that while the e-health tool presented here is intended for operational use, these algorithms have not yet been used by the existing CHW program in Ifanadiana District or integrated into CHWs' workflows. For this, implementation research is needed in order to tailor the general method described here to users' expectations, field constraints, particular CHW program features, or use with existing software. While this was beyond the scope of this study, such work will ensure that the tool contributes to decision making and program implementation of the local community health program.

## Results

### Planning resources for mass distribution campaigns

To cover the 108,000 buildings in Ifanadiana district during mass distribution campaigns (both occupied households and unoccupied buildings), a total of 4,639 person-days would be required to complete the work. These personnel would need to travel over 44,000 km and work for nearly 35,000 hours to successfully complete the work across the district, with important differences between fokontany (Table 2) and between communes (S1 Table). The required personnel-days ranged from 5 to 77 per fokontany. Nearly one third of fokontany required less than 15 personnel-days to cover the whole catchment, whereas over one quarter of fokontany (27.7%) required over 30 personnel-days (Table 2). Larger fokontany with more dispersed households required a larger number of personnel compared to smaller communities with residential zones in close proximity (Fig 2A). Travel time between buildings represented on average 25% of total work time. There was a strong significant correlation between personnel requirements and the number of buildings per fokontany (pearson's $\rho = 0.96$), while this association was less strong for the total catchment area ($\rho = 0.56$) and the population size ($\rho = 0.52$) (S2 Fig).

### Planning resources for proactive community health programs (proCCM)

To cover the estimated 42,481 households to be visited during proCCM programs in Ifanadiana District, a total of 1,508 person-days would be required to complete one visit per

**Table 2. Personnel requirements per fokontany for mass distribution campaigns.**

| Personnel-days required | Number of fokontany (%) | Average per fokontany | | | |
| --- | --- | --- | --- | --- | --- |
| | | Buildings | Travel distance (travel time) | Work time | Personnel—days |
| [0,15] | 64 (32.8%) | 275 | 63km (13h) | 73h | 11 |
| (15,30] | 77 (39.5%) | 509 | 189km (38h) | 161h | 22 |
| (30,45] | 37 (19.0%) | 832 | 382km (76h) | 278h | 37 |
| (45,60] | 11 (5.6%) | 1086 | 602km (120h) | 384h | 51 |
| > 60 | 6 (3.1%) | 1401 | 860km (172h) | 512h | 67 |
| *Total district* | *195 (100%)* | *107,928* | *44,496km (8,899h)* | *34,674h* | *4,639* |

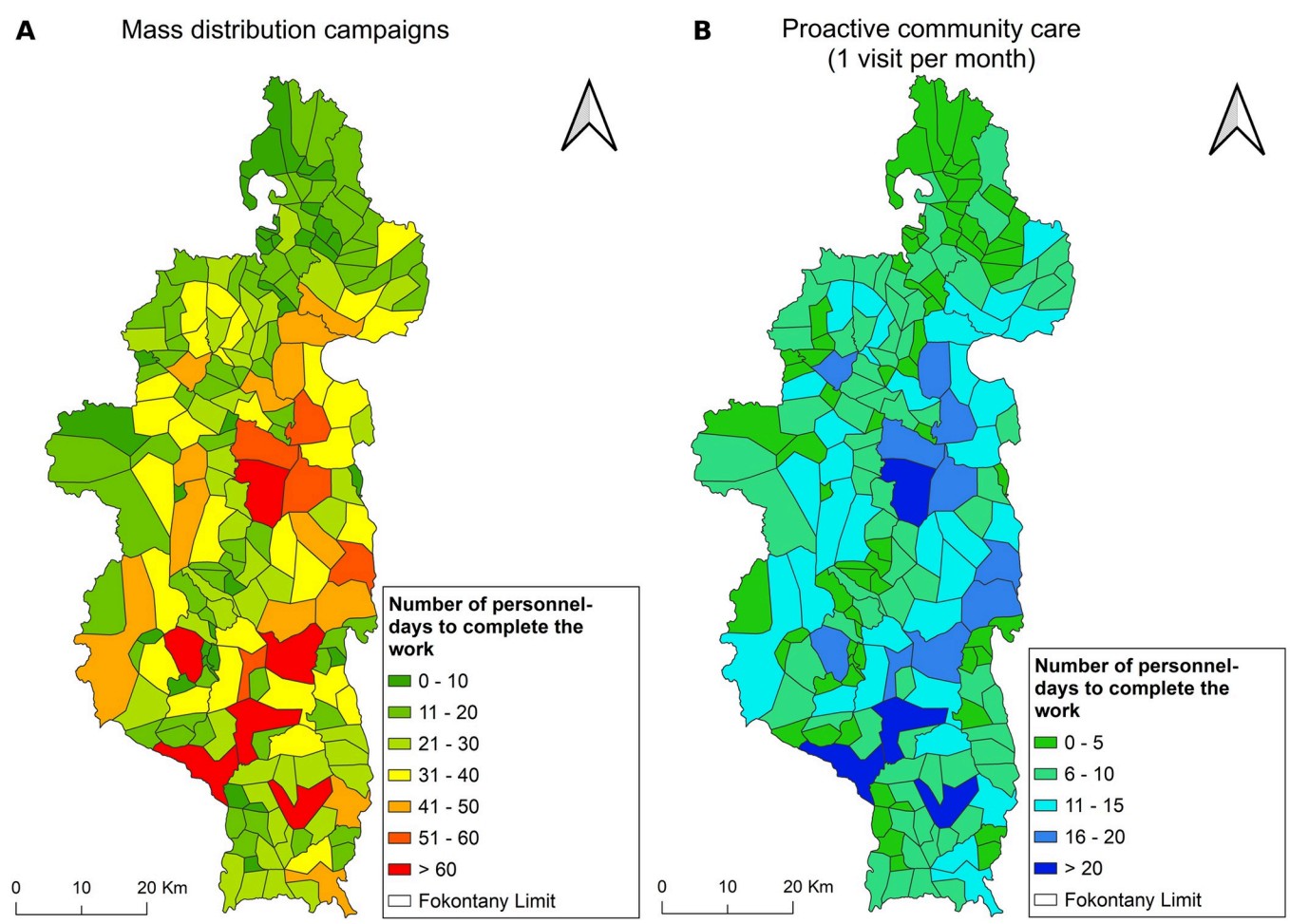

**Fig 2. Spatial distribution of personnel-days requirements per fokontany to deliver community health interventions in Ifanadiana district.** A) Mass distribution campaigns. B) Proactive community case management programs (one visit per month). Sources: HumData Madagascar (https://data.humdata.org/dataset/cod-ab-mdg) used for administrative boundaries.

household per month. As with mass distribution programs, requirements for proCCM programs varied widely across fokontany (Table 3) and communes (S2 Table) (Fig 2B). While three quarters of fokontany would only require one CHW working for up to 2 weeks per month (0–10 personnel-days), four fokontany (2.1%) would require more than one full time

**Table 3. Personnel requirements per fokontany for proCCM programs.**

| Personnel-days required per visit | Number of fokontany (%) | Households | Travel distance (travel time) | Work time | Personnel–days required | | |
| --- | --- | --- | --- | --- | --- | --- | --- |
| | | | | | 1 visit per month | 2 visits per month | 4 visits per month |
| [0,5] | 74 (38.0%) | 109 | 32km (6h) | 24h | 4 | 8 | 15 |
| (5,10] | 73 (37.4%) | 216 | 80km (16h) | 51h | 8 | 15 | 30 |
| (10,15] | 34 (17.4%) | 347 | 155km (31h) | 86h | 13 | 25 | 50 |
| (15,20] | 10 (5.1%) | 456 | 251km (50h) | 123h | 18 | 34 | 69 |
| > 20 | 4 (2.1%) | 580 | 355km (71h) | 163h | 23 | 45 | 89 |
| Total district | 195 (100%) | 42,481 | 17,429km (3,486h) | 10,259h | 1,508 | 3,016 | 6,032 |

*The header row "Average per fokontany" spans the Households, Travel distance, Work time and Personnel–days required columns.*

CHW, as it exceeded 20 personnel-days of work (Table 3). Travel time between households represented 34% of total work time. As expected, personnel-day requirements increase proportionally with the increasing frequency of visits (Table 3), but the number of CHWs required to meet this visit frequency was also determined by the limit of 20 workdays per month, with some fokontany able to meet higher visit frequencies with fewer personnel. These results were robust to the distribution of occupied households within the full building dataset, as demonstrated by a sensitivity analysis of 100 randomizations of household locations. At the fokontany scale, the greatest variation was seen in fokontany requiring between 10–15 personnel days, with a difference of up to 2 personnel-days across randomizations, although the majority of fokontany (135) experienced no difference between the initial estimate and the estimates during the resampling (S3 Fig, S3 Table). When aggregated across the district, the total number of personnel-days required ranged from 1490–1520 across the randomizations, less than a 0.02% difference from the initial estimate (S4 Fig).

Given that proCCM programs can vary in terms of frequency and activities to be implemented, which influences the average time spent per household, we explored the resources needed under a range of program design scenarios. The estimated CHWs needed in Ifanadiana district to implement a proCCM program with one visit per month to households ranged from 199 CHWs when visits with and without a sick child were held at the minimal visit times to 578 CHWs when assuming maximal visit times (Fig 3). This range increased to a minimum of 202 and a maximum of 1060 CHWs needed for 2 visits per month, and to a minimum of 268 and a maximum of 2020 for 4 visits per month (Fig 3).

## Optimizing programmatic activities with the VRPTW routing algorithm

By combining the output of the VRPTW algorithm (order of buildings to visit each day) with the GIS information on shortest paths and household locations from OSRM, we were able to obtain and visualize optimal daily schedules and itineraries for CHWs to conduct door-to-door interventions and cover all the households in their fokontany (Fig 4). Because the ultimate goal was to support decision-making in the field, we developed an e-health tool that facilitates the use of optimization results by community health programs. This tool provides an estimation of resources needed for each intervention and catchment area, as well as visualization of schedules and itineraries for each CHW (Fig 5). It also allows programs to simulate more tailored personnel needs according to the particular design and assumptions of an intervention, varying the duration of visits and the number of visits to be conducted (Fig 5, left side panel). Due to processing limitations, these parameters were defined in advance, with ranges similar to those in Fig 3, and pre-run prior for display within the dashboard. Results can be aggregated by district and commune, with details available at the fokontany level. Daily itineraries can be visualized over a map of transportation routes or high resolution satellite imagery available through *OpenStreetMap*. Besides visualization online, data can be exported in GPX, PDF, and CSV formats to maintain data interoperability, facilitating integration with other tools and use offline. GPX downloads allow geographical information (routes and household locations) to be saved for use on tablets or smartphones, PDFs can be used for printing paper versions of the plans, and CSV files provide a textual breakdown of routes along with the defined visit times for each household. The routing optimization tool is accessible via the following link: https://research.pivot-dashboard.org/.

## Discussion

Community health programs and last mile delivery interventions are gaining increasing relevance in international efforts to achieve universal health coverage and improve population

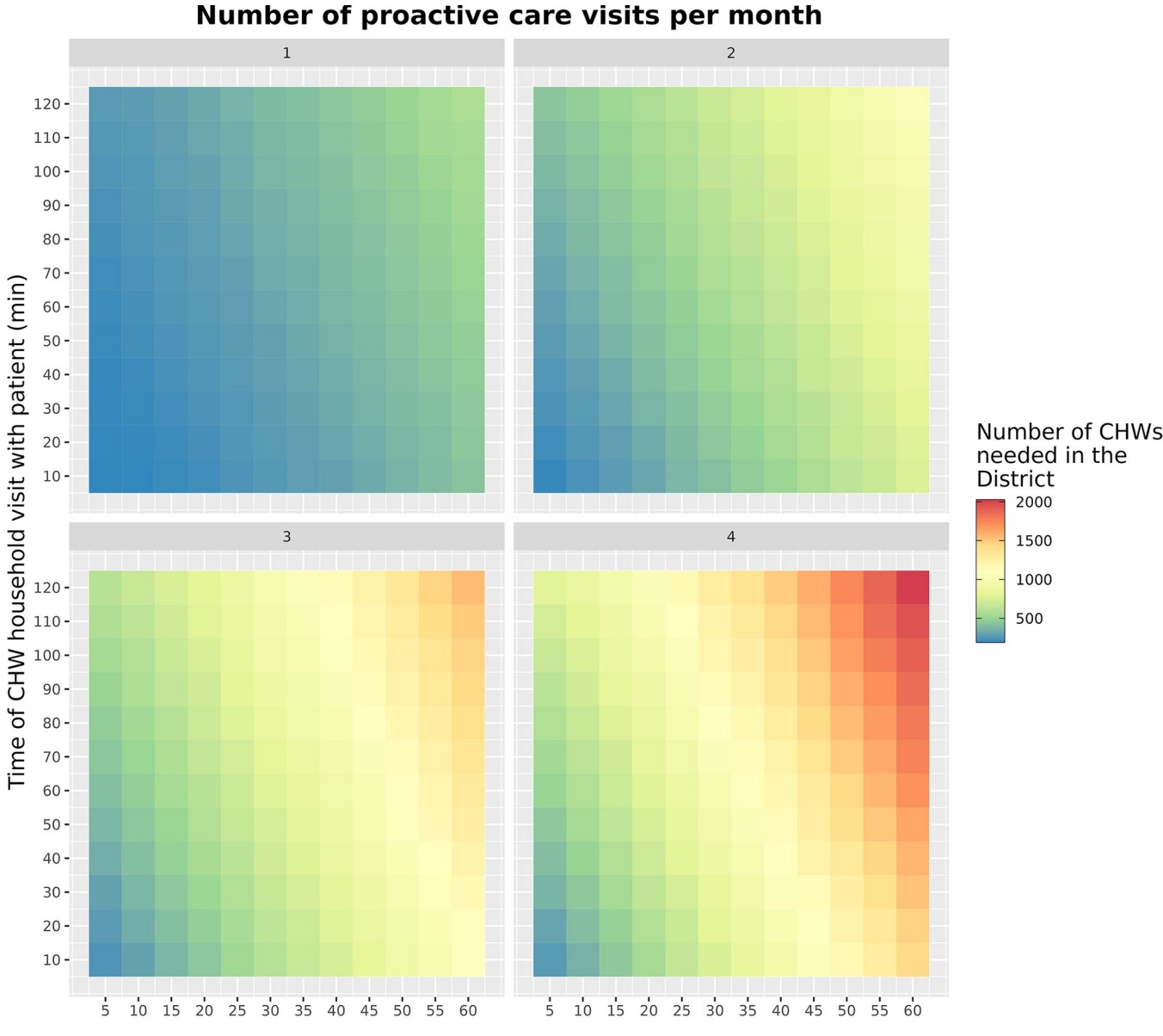

**Fig 3. Number of CHWs required for proCCM programs in Ifanadiana District under a range of program design scenarios.** The number of CHWs needed were obtained for each combination of three parameters used in the algorithm: number of visits per month, the visit time in households without patients, and the visit time in households with patients.

health in low resource settings [3]. As a result, there is a corresponding interest in optimizing different aspects of program design and implementation to achieve the best possible outcomes at the community level [3]. CHWs are often involved in door-to-door delivery of medical products and services that require covering large geographical areas, such as during proCCM programs and mass distribution campaigns. These interventions represent a unique opportunity for integration with geographic optimization tools to inform their design and implementation, which is commonplace in many other delivery sectors [26]. Here, we combined a complete mapping of a rural district of Madagascar on *OpenStreetMap* with optimization

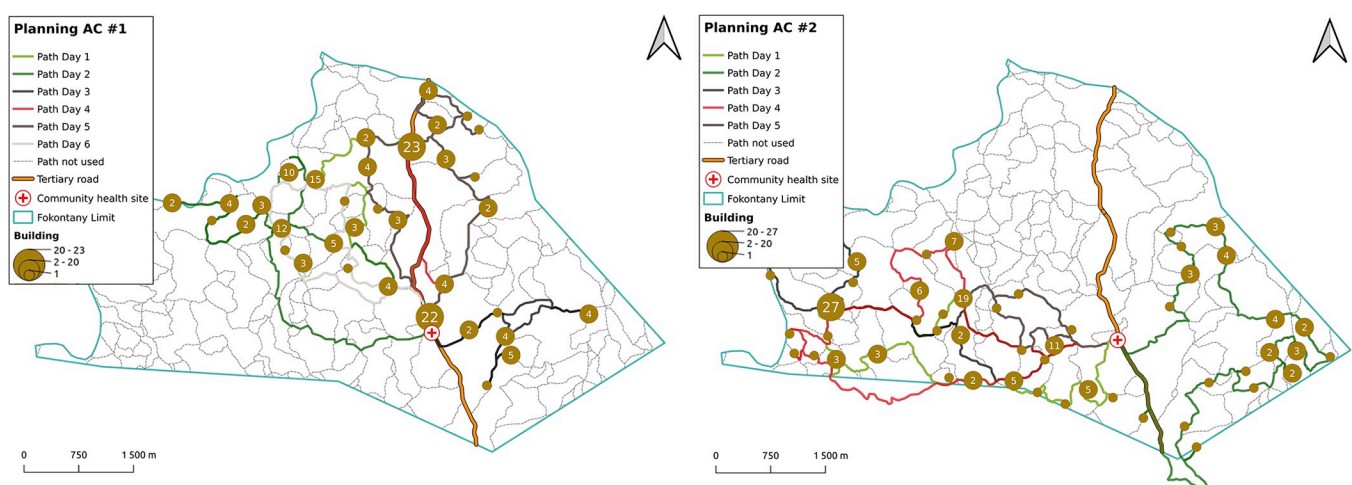

**Fig 4. Illustrative example of optimal daily itineraries for two CHWs to visit every household in their catchment.** Output includes i) the routes CHWs need to take, with different colors for each day, ii) the number of households to visit each day in each location, with brown circles proportional to the number of households to be visited, iii) unused roads and paths, and iv) CHW catchment boundaries and location of the CHS. Fokontany limits determine the households located in the CHW catchment, but paths outside the fokontany can be used during daily itineraries if they represent the shortest route, as illustrated in the right map. Sources: HumData Madagascar (https://data.humdata.org/dataset/cod-ab-mdg) used for administrative boundaries, Contains information from OpenStreetMap and OpenStreetMap Foundation, which is made available under the Open Database License (https://www.openstreetmap.org/copyright).

algorithms based on the Vehicle Routing Problem with Time Windows (VRPTW) to inform the planning and implementation of community health interventions requiring door-to-door delivery. We found that the resources necessary to implement these interventions in different CHW catchments within a health district could vary by over fivefold for the same intervention

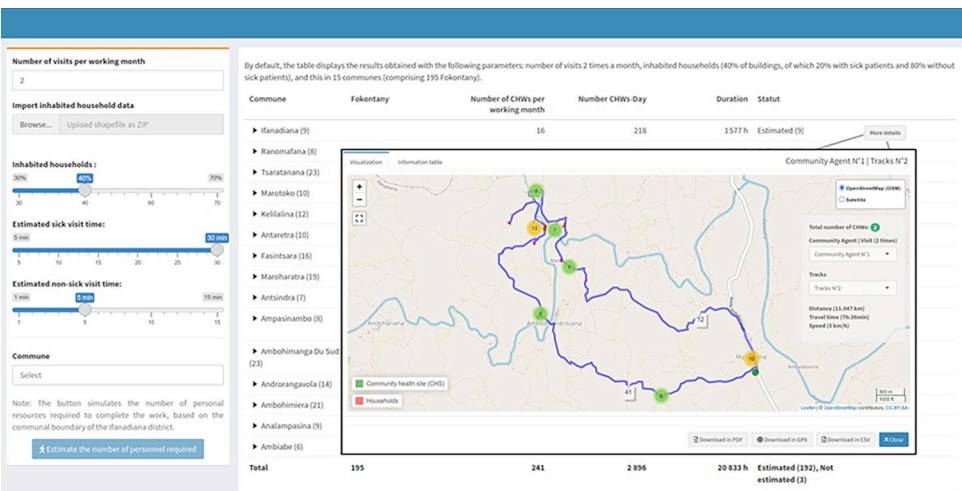

**Fig 5. Routing optimization tool implemented as an interactive dashboard for use by community health programs in Ifanadiana District.** Screenshot displays one of several functionalities of this tool. The map at the center showcases the detailed daily route of a CHW for one day, including the households to be visited, arranged in ascending order until returning to the CHS. Parameters of the algorithm (e.g. visit times, number of monthly visits) can be changed via the left-hand side panel and key output information (number of personnel-days, etc.) is provided for each fokontany and Commune in the table underneath the map. The dashboard is publicly available at https://research.pivot-dashboard.org/, under the "Community resource planning" tab. Sources: Base map and data from OpenStreetMap and OpenStreetMap Foundation. (https://www.openstreetmap.org/copyright).

due to differences in catchment geography and population (Tables 2 and 3), and by nearly ten-fold for the same catchment due to differences in program design (Fig 3).

Our approach, which allows program managers to estimate resources necessary for various intervention scenarios and recommend optimal itineraries for CHWs to conduct household visits in their catchment via an e-health tool for operational use, aims to address key challenges often identified by implementers of last mile delivery interventions in low resource settings. Indeed, the misestimation of the number of personnel involved or drugs procured, and the difficulty of ensuring distribution for hard to reach populations are barriers that can lead to insufficient coverage during mass drug administration campaigns [34,35]. Similar challenges are faced during mass distribution campaigns of insecticide-treated bed nets, where incomplete mapping of target populations due to resource constraints, difficulties in transportation and low distribution rates in rural areas have been identified as key factors explaining low coverage rates at the population level [36,37]. For childhood vaccinations, immunization coverage is typically lowest in the most remote communities, despite mass campaigns and other outreach activities [14,38]. ProCCM programs are rarer, and less is known about barriers to their implementation. However, in the settings where proCCM has been implemented, it has been suggested to reduce neonatal deaths and improve the coverage of maternal and child care services [4–6,39,40]. It is becoming standardized for proactive community health workers to use smartphone applications to assist with consultations during home visits [41], and there is evidence that adding personalized performance dashboards can improve CHW productivity [42]. Adding routing algorithms such as those proposed here to assist CHWs and program managers in optimizing daily schedules and itineraries based on the geographic context of the catchment area could therefore contribute to additional increases in productivity and population coverage.

With rapid advances in big data analytics, particularly represented by the enthusiasm around artificial intelligence, the basic challenges remain for households at the last mile whose growing disease burdens are met with persistent barriers to care. For such advances in analytics to have impact, they must be anchored to high quality, highly granular data, and linked to services that can meaningfully use such data to inform programs. One powerful direction is the navigation of geography itself: optimization of travel routes has not only revolutionized home delivery services, supply chain management, and even domestic healthcare, but has touched nearly every major industry in high income countries. In the health sector, route optimization approaches have been primarily applied for home healthcare in developed countries, to help optimize the logistics and scheduling of home visits by nurses and doctors [43–45]. Many variations exist to adapt the base algorithms to the specificities of each program, such as client and workforce preferences, or constraints in terms of qualification level for health personnel or vehicle fleet available [16,46]. For community health programs in developing countries, efforts have tended to focus on optimization of CHS locations to be closest to the populations they serve [13,47], but more sophisticated applications for door-to-door delivery have lagged behind. Here, we expand the use of VRPTW algorithms to community health delivery in rural areas, incorporating a comprehensive geographic data system of footpaths and buildings. For proCCM programs, we considered a simple scenario where CHWs visit households on a frequent and fixed basis, but are not required to carry follow-up visits to certain patients with a different periodicity or priority level. Although this is the case of proCCM interventions implemented in settings such as Senegal and Madagascar [11,48], where the scope was limited to malaria, other proCCM programs require a larger set of tasks and therefore more complex scheduling [6,15]. Future research can build on the work presented here to integrate additional context-specific constraints to support the implementation of such programs.

The innovation of our approach is the leveraging of an exhaustive dataset of all the roads, footpaths and buildings in a rural health district collected via *OpenStreetMap* mapping prior to integration with VRTPW algorithms for use by community health programs. Although the approach is flexible and can be adapted to a large number of community health delivery interventions, the main bottleneck in its application elsewhere is the availability of similar high-quality GIS datasets. The completeness of global public maps such as OSM is generally low in rural, low resource settings, but mapping efforts such as those facilitated by the Humanitarian *OpenStreetMap* Team (HOT) are progressively closing the gap for certain regions [18]. For instance, urban OSM building completeness was higher in sub-Saharan Africa than in any other world region except for North America, Europe and Central Asia [18]. In addition, while mapping on OSM is typically done by a community of mappers and can be time consuming, which explains the slow progress in OSM completeness, recent years have seen the rise of multiple tools and datasets powered by artificial intelligence. For instance, both Microsoft and Google have released global building datasets and Meta developed an AI assisted editor to add map features generated by AI to OSM, including roads and buildings [49]. As these tools improve and expand, high-quality GIS datasets will soon be available for many other rural, low resource settings, enabling the scale-up of our approach to inform community health delivery in other settings.

Our study had several limitations. First, we did not account for differences in travel speed according to weather conditions, seasonal factors, landscape or elevation slopes, all of which are known to influence travel and commute times [19,28]. Although walking speed used here corresponded to average estimates from fieldwork done with CHWs [19], this omission could have led to slight underestimations or overestimations in travel time which could have influenced estimates of required personnel for both interventions. Second, the simulations in the study rely on *a priori* assumptions made for several parameters in the algorithm, which were based on existing data for the district and discussions with local community health program managers. Differences between assumed values and real world implementation could introduce uncertainty or biases in our estimates. We partly addressed this by simulating different scenarios with a range of values for some key parameters such as visit times. In addition, we retrospectively compared our predictions with actual personnel deployed in the field during a census survey conducted in two communes, prior to implementation of proCCM. This comparison revealed that the optimization algorithm predicted a number of person-days 21%-29% lower than those deployed in each commune, in line with our hypothesis that optimized schedules would result in lower resource needs (S1 Text). Third, while the algorithm for proCCM as presented here, with a random allocation of buildings as inhabited or uninhabited households, can be used to estimate human resources needed during the planning phase, the actual itineraries were illustrative and not for implementation purposes. In practice, use of scheduling and itineraries for proCCM would require conducting a census of the CHW catchment in order to integrate the true location of inhabited households into the algorithm. However, a sensitivity analysis revealed that the estimates of personnel-days required were remarkably robust to the distribution of inhabited and uninhabited buildings, suggesting that human resource estimates, at least, may not require these exact locations. Finally, the choice of routes and order of households to be visited can depend on factors other than what is most optimal geographically, such as CHW preferences, household availability, or insecurity and inaccessibility in certain areas. Operational research is needed to assess the acceptability of these tools by CHWs and program managers and to adapt the algorithms according to their feedback and preferences.

More broadly, while this exercise demonstrates the potential to optimize community health programs and interventions, it also highlights aspects of community healthcare that are less amenable to algorithmic optimization. The recent developments in big data analytics, artificial

intelligence (AI) and availability of novel data streams have increased the interest in and opportunities for their use to improve public health processes [50], while raising questions of equity and bias [51]. In our case, the VRTPW algorithm can accurately plan interventions that are predictable and repetitive, such as a MDA program, a census, or a simple proCCM intervention (e.g. proactive detection and treatment of common childhood illnesses). However, much of a CHW's duties can involve reacting to health needs as they arise, and the factors that influence variation in their day-to-day activities are not reliably captured in ways that can be predicted in advance. For instance, when we retrospectively compared our predictions with actual personnel deployed in the field during a census survey conducted in two communes, prior to implementation of proCCM, results revealed that the optimization algorithm predicted a number of person-days 21%-29% lower than those deployed in each commune, in line with our hypothesis that optimized schedules would result in lower resource needs (S1 Text). However, when compared to a pilot proCCM program that has taken place in one commune since October 2019, the VRPTW algorithm predicted personnel needs substantially lower in the standard scenario (Fig 2B, Table 3) than those actually used in the intervention. While part of this discrepancy is due to the more efficient movement of CHWs via the VRTPW optimization process, it is also due to the VRTPW's inability to capture the daily changing tasks of a CHW and other real-life factors when applied at a district-level. For example, CHWs during the pilot proCCM program were charged with follow-up of tuberculosis and malnutrition patients, visiting some patients daily, or following up on others based on the terms of their discharge, a task that was not taken into account during this theoretical exercise. In instances where CHWs' responsibilities may change daily, a more customized e-Health tool that combines patient-level data with geographic data is necessary to provide CHW's with their daily routes. While geographic optimization can help CHWs with route planning, it remains just one piece of a larger puzzle to improve community health programs' ability to serve the population.

## Conclusion

This study demonstrates an innovative approach to inform the planning and implementation of door-to-door delivery activities at the community level that relies on existing geographic optimization methods, by combining a Vehicle Routing Problem with Time Windows (VRPTW) algorithm with an exhaustive GIS dataset consisting of all road networks and buildings in a rural health district, obtained through *OpenStreetMap* mapping. The algorithm, which predicts the number of personnel-days needed to implement these activities according to program design and the characteristics of the CHW catchment, can also generate detailed schedules and itineraries for implementation by CHWs and field teams via a dedicated "e-health" platform. Implementation of this tool into a community health program will require a participatory approach, such as community-based participatory research or participatory action research (PAR) [52], to engage CHWs and other end-users in the refinement and deployment of the application in the field and to ensure its validity among CHWs. Indeed, similar approaches have been used to identify obstacles to the adoption of mHealth tools and to co-design [53] them in a rural health context. The tool could also be integrated into existing mobile health tools used by CHWs, such as comm-Care, to encourage its uptake. While implementation research is needed to effectively integrate these tools for real world program implementation, their scale up could help optimize community health programs in line with WHO recommendations, taking advantage of rapid advances in quality and the quantity of data and analytics from around the world to inform the most urgent programs at the last mile.

## Supporting information

**S1 Fig. Workflow for VRPTW algorithm implementation.**
(PDF)

**S2 Fig. Correlation between daily personnel requirements and CHW catchment characteristics in Ifanadiana district.** A-C: Mass administration campaigns, D-F: Proactive Community Health (one visit per month). Panels A and D plot a linear regression line due to the nature of the relationship, while the other panels use non-linear local regression fitting.
(PDF)

**S3 Fig. Distribution of the median difference in personnel-days required to conduct proCCM across 100 randomizations of household occupancy status and the initial estimate reported in the main text by fokontany (n = 195).**
(PDF)

**S4 Fig. Distribution of the difference in personnel-days required to conduct proCCM across the whole district for each resampling of household occupancy status (n = 100) and the initial estimate reported in the main text (1,508 personnel-days).**
(PDF)

**S1 Table. Estimated number of personnel required per commune for mass distribution campaigns.**
(PDF)

**S2 Table. Estimated number of personnel required per commune for proCCM programs.**
(PDF)

**S3 Table. Comparison in personnel days required per fokontany for proCCM programs between 100 randizations (resampling) of household occupancy status and the initial estimate in the manuscript, assuming one visit per month and default parameters.**
(PDF)

**S1 Text. Comparison of algorithm predictions and field-observed CHW needs in Ifanadiana district.**
(PDF)

## Acknowledgments

We would like to thank Vincent Herbreteau, Christophe Revillion and Jérémy Commins for their involvement in the *OpenStreetMap* mapping, as well as members from the Pivot Community team for their inputs on community health activities in Ifanadiana District.

## Author Contributions

**Conceptualization:** Mauricianot Randriamihaja, Andres Garchitorena.

**Data curation:** Mauricianot Randriamihaja, Felana Angella Ihantamalala, Feno H. Rafenoarimalala, Benedicte Razafinjato, Andres Garchitorena.

**Formal analysis:** Mauricianot Randriamihaja, Felana Angella Ihantamalala, Michelle V. Evans.

**Funding acquisition:** Andres Garchitorena.

**Investigation:** Mauricianot Randriamihaja, Feno H. Rafenoarimalala, Karen E. Finnegan, Matthew H. Bonds.

**Methodology:** Mauricianot Randriamihaja, Felana Angella Ihantamalala, Michelle V. Evans.

**Project administration:** Andres Garchitorena.

**Supervision:** Matthew H. Bonds, Michelle V. Evans, Andres Garchitorena.

**Validation:** Mauricianot Randriamihaja, Feno H. Rafenoarimalala, Karen E. Finnegan, Luc Rakotonirina, Benedicte Razafinjato.

**Visualization:** Mauricianot Randriamihaja.

**Writing – original draft:** Mauricianot Randriamihaja, Michelle V. Evans, Andres Garchitorena.

**Writing – review & editing:** Mauricianot Randriamihaja, Felana Angella Ihantamalala, Feno H. Rafenoarimalala, Karen E. Finnegan, Luc Rakotonirina, Benedicte Razafinjato, Matthew H. Bonds, Michelle V. Evans, Andres Garchitorena.

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
