## [Decision Letter · Decision Letter 0]

24 Jun 2024

PDIG-D-24-00099

Combining participatory mapping and route optimization algorithms to inform the delivery of community health interventions at the last mile

PLOS Digital Health

Dear Dr. Randriamihaja,

Thank you for submitting your manuscript to PLOS Digital Health. After careful consideration, we feel that it has merit but does not fully meet PLOS Digital Health's publication criteria as it currently stands. Therefore, we invite you to submit a revised version of the manuscript that addresses the points raised during the review process.

Please submit your revised manuscript within 60 days Aug 23 2024 11:59PM. If you will need more time than this to complete your revisions, please reply to this message or contact the journal office at digitalhealth@plos.org. Please include the following items when submitting your revised manuscript:

We look forward to receiving your revised manuscript.

Kind regards,

Matthew O. Wiens

Academic Editor

PLOS Digital Health

Journal Requirements:

Additional Editor Comments (if provided):

The reviews point out several key areas that should be addressed in a revision. Futhermore, I believe the discussion could be strenghened by including some additinal content relating to how these findings might be best incorporated into existing systems, or perhaps which strategies could be employed to aid in the future adoption of geographic optimization algorithms into health systems.

Reviewers' comments:

Reviewer's Responses to Questions

**Comments to the Author**

1. Does this manuscript meet PLOS Digital Health’s publication criteria? Is the manuscript technically sound, and do the data support the conclusions? The manuscript must describe methodologically and ethically rigorous research with conclusions that are appropriately drawn based on the data presented.

Reviewer #1: Yes

Reviewer #2: Yes

Reviewer #3: Partly

2. Has the statistical analysis been performed appropriately and rigorously?

Reviewer #1: Yes

Reviewer #2: Yes

Reviewer #3: No

3. Have the authors made all data underlying the findings in their manuscript fully available (please refer to the Data Availability Statement at the start of the manuscript PDF file)?

Reviewer #1: Yes

Reviewer #2: Yes

Reviewer #3: Yes

4. Is the manuscript presented in an intelligible fashion and written in standard English?

Reviewer #1: Yes

Reviewer #2: Yes

Reviewer #3: Yes

5. Review Comments to the Author

Reviewer #1: This article describes the methodology to inform delivery of Community health interventions through the optimization of the routing of CHW households visits. It uses a combination of OSM data and a routing algorithm to provide optimum numbers and routes for CHWs according to two types of community health interventions. 

The article is overall very well written with sound discussion and conclusion, and with well-made maps and Figures. It is a timely addition to the current literature on using high-resolution geospatial data to optimize the provision of health services. There are several points the need to be clarified, especially in the Methods section, before the manuscript can be accepted:

- Line 156: you need to make it more explicit what you mean by unoccupied households, because it could be understood as being empty because at the time of the visit the households members are out and not available. But I understand that your assumption of 40% occupancy, given the overall population, means that 60% of the building are something else than houses else (e.g. a storage house). It would be important to state it and explain why the digitalization exercise cannot discriminate the building function.

- Regarding the assumed 40% occupancy rate of the building, and the random attribution of that status to all buildings, I would have liked to see a sensitivity/uncertainty analysis on the impact of the randomization of the building attribution. Would another randomization significantly change the estimation of the number of required CHW? I suspect not, but at least it would be good to discuss it in the Discussion section, with a possible justification. 

- Lines 234-237: it is great that this e-health tool is available for users. I suggest you make it explicit here that when the user changes some of the (default) simulation parameters, there is no re-run of the routing and optimization analyses, but simply a display of the pre-run set of analyses (if I understood correctly, but if analyses are ran again, you need to better explain it).

- Figure 4: I was nicely surprised to see in the right panel that the routing allows for part of the route to go through neighboring fokontany. This would be worth mentioning, as some other tools might consider fokontany borders to be strict barriers, unrealistically.

- Lines 398-400: you state that your hypothesis was that optimized schedules would results in lower resources needs. This lower resource needs seems to be indeed confirmed by your data in the field study presented in S1 text. However, how can you be sure that these results do not merely reflect the fact that your simulations underestimate the travel time and the duration of the visits, due to the many real-life factors that would lengthen travel and visit times? Could your data provided in S1 text be rather used to calibrate your simulation parameters ? 

- Lines 417-419: are these line somehow redundant with what you say on liens 398-400 ? if so you should merge the discussion bits on the predicted lower personnel needs.

Reviewer #2: The authors address a crucial issue in health service delivery: reaching the last mile, which involves providing healthcare to the most inaccessible areas. Their study focuses on a rural region in Madagascar, where they have applied an existing method in a novel context. Typically, the lack of comprehensive road networks and geographic data in low- and middle-income countries (LMICs) hinders the implementation of such models. However, continuous efforts to collect and complete this data have resulted in highly detailed spatial layers, enabling the authors to successfully run the VRPTW algorithm.

The article is well-written, thorough, and has already demonstrated its utility and applicability in the field. I recommend that the authors address a few minor issues to further improve the article.

Feedback:

-I wonder why the authors have chosen Fokontany as catchment areas and have not defined these by travel time or distance? Is this how the local government decides on health service areas? I recommend the authors to add a justification for this decision.

-In addition, did the authors allow CHWs in their model to cross Fokontany borders or are the borders considered as barriers? From the map in Figure 1C it seems that some households may be closer to CHWs from the neighboring Fokontany/catchment area. Has this been included in the model? I would recommend the authors to clarify this. 

-Line 157-159: “We randomly assigned a building’s status (e.g. occupied vs. unoccupied) following the percentages above for each fokontany and ensured there was no spatial correlation in this sampling procedure. The scenarios differed primarily in the percentage of households to be visited, the frequency of visits, and the duration of each visit.” Could this be cross checked with results from high resolution HRSL building footprint data or WorldPop building data?

-The authors choose a walking speeds of 5km/h and mention in their limitations this may be an under- or overestimation. Research in different settings has shown that travel speeds on different terrain types than tarmac roads can be lower especially considering the altitude differences in the fokonotany of interest. I would suggest the authors to further clarify their decision and to have a look at the important work done by Watmough et al (https://www.nature.com/articles/s41597-022-01274-w).

-Line 251-252: “To cover the 108,000 buildings in Ifanadiana district during mass distribution campaigns, a total of 4,639 person-days would be required to complete the work.” Clarify whether these are inhabited buildings or all buildings. 

-Line 256-258: “Fokontany with larger catchment areas and more dispersed households required a larger number of personnel compared to smaller communities with residential zones in close proximity (Figure 2A).” This sentence seems a little strange to me, because the catchment areas are the same as the boundaries of the fokontany. Suggestion to change to: “Larger fokontany with more dispersed households…”

-It is unclear whether the some of the building status will be fed back into the models. If CHWs visit a building that is uninhabited, will the model be updated to present a more realistic assumption rather than a random classification?

Reviewer #3: The concept behind this paper is great, and I'm glad to see an innovative route optimization approach to enhance equitable care coverage by community health workers. Below are some of my comments:

Overall: Firstly, the authors need to provide more concise background information on current coverage of CHWs in the districts. There is a mention of 180 CHWs supported by NGO, but what about government CHWs, combined what does the current coverge looks like. How many people are currently not being reached? Such information is critical to justify for your innovation. Also, the authors didn’t provide the specifics of complex CHW program (line 51-52), which is a bit vague. Although. , there is mention of If the tool is intended for program managers, as indicated on line 228, more information is needed about these users and their capacity to utilize this application. The software used in this research may not be easily transferable to community health worker supervisors or district health teams. Include a justification regarding the intended target user and its relevance.

Your title indicated participatory mapping, but I failed to find how this was executed. What aspect of this research involved participatory methods? If validation of routes was done by two technical people, that seems more like a research team effort than community participation. For true participatory mapping, involving community members to validate digitized roads, buildings, etc., sourced from OSM, or including CHWs, is necessary. The participatory mapping aspect needs thorough and clear description.

Another major flaw concerns the specifics of route optimization, particularly the random selection of uninhabited buildings. If a participatory approach was used, sampling a few buildings in-person to validate/ground truth your digitization could improve the credibility, the current approach is not sufficient as a reliable node in the optimization process. And Solely relying on spatial correlation checks does not sufficiently address limitations in route selection from CHWs. The core of this paper revolves around finding the best route from CHWs to households. This is where a participatory approach could be instrumental. Also, why use a vehicle route optimization approach when the primary mode of transport for CHWs is walking in hard-to-reach areas? Have you explored other route optimization or coverage mapping exercises?

In the abstract, you mention the feasibility and utility of the tool; however, the conclusion does not adequately address these intended goals.

The background could be simplified by structuring it into introductory paragraphs:

• What is the problem

• What do we know about it

• What do we not know about it

• Why we did you conduct this study and why it is relevant locally or in other similar settings?

Specifically: 

Background on how the optimization tool works, even, if possible, visualize? And explain the process of adaptation for your setting. 

Also, there are few places where the importance of the tool and approach is repeated in the intro, methods, results and discussion. 

Methods and results are partially merged, please discern the two sections.

6. PLOS authors have the option to publish the peer review history of their article (what does this mean?). If published, this will include your full peer review and any attached files.

**Do you want your identity to be public for this peer review?** For information about this choice, including consent withdrawal, please see our Privacy Policy.

Reviewer #1: No

Reviewer #2: No

Reviewer #3: No

---

## [Decision Letter · Decision Letter 1]

20 Aug 2024

Combining OpenStreetMap mapping and route optimization algorithms to inform the delivery of community health interventions at the last mile

PDIG-D-24-00099R1

Dear Randriamihaja,

We are pleased to inform you that your manuscript 'Combining OpenStreetMap mapping and route optimization algorithms to inform the delivery of community health interventions at the last mile' has been provisionally accepted for publication in PLOS Digital Health.

Best regards,

Matthew O. Wiens

Academic Editor

PLOS Digital Health

Reviewer Comments (if any, and for reference):

Reviewer's Responses to Questions

**Comments to the Author**

1. If the authors have adequately addressed your comments raised in a previous round of review and you feel that this manuscript is now acceptable for publication, you may indicate that here to bypass the “Comments to the Author” section, enter your conflict of interest statement in the “Confidential to Editor” section, and submit your "Accept" recommendation.

Reviewer #1: All comments have been addressed

Reviewer #2: All comments have been addressed

Reviewer #3: All comments have been addressed

2. Does this manuscript meet PLOS Digital Health’s publication criteria? Is the manuscript technically sound, and do the data support the conclusions? The manuscript must describe methodologically and ethically rigorous research with conclusions that are appropriately drawn based on the data presented.

Reviewer #1: Yes

Reviewer #2: Yes

Reviewer #3: Yes

3. Has the statistical analysis been performed appropriately and rigorously?

Reviewer #1: Yes

Reviewer #2: Yes

Reviewer #3: I don't know

4. Have the authors made all data underlying the findings in their manuscript fully available (please refer to the Data Availability Statement at the start of the manuscript PDF file)?

Reviewer #1: Yes

Reviewer #2: Yes

Reviewer #3: Yes

5. Is the manuscript presented in an intelligible fashion and written in standard English?

Reviewer #1: Yes

Reviewer #2: Yes

Reviewer #3: Yes

6. Review Comments to the Author

Reviewer #1: (No Response)

Reviewer #2: The authors have appropriately addressed all my comments from review round 1. I have no further reviews for the authors.

Reviewer #3: (No Response)

7. PLOS authors have the option to publish the peer review history of their article (what does this mean?). If published, this will include your full peer review and any attached files.

**Do you want your identity to be public for this peer review?** For information about this choice, including consent withdrawal, please see our Privacy Policy.

Reviewer #1: **Yes: **Nicolas Ray

Reviewer #2: **Yes: **Fleur Hierink

Reviewer #3: No
